# Polyphenols Sourced from *Ilex latifolia* Thunb. Relieve Intestinal Injury via Modulating Ferroptosis in Weanling Piglets under Oxidative Stress

**DOI:** 10.3390/antiox11050966

**Published:** 2022-05-13

**Authors:** Xiao Xu, Yu Wei, Hongwei Hua, Xiaoqing Jing, Huiling Zhu, Kan Xiao, Jiangchao Zhao, Yulan Liu

**Affiliations:** 1Hubei Key Laboratory of Animal Nutrition and Feed Science, School of Animal Science and Nutritional Engineering, Wuhan Polytechnic University, Wuhan 430023, China; xuxiao@whpu.edu.cn (X.X.); yuwei20000306@163.com (Y.W.); hw-hua@hotmail.com (H.H.); xiaoqing819903@163.com (X.J.); zhuhuiling@whpu.edu.cn (H.Z.); canxiaok@whpu.edu.cn (K.X.); jzhao77@uark.edu (J.Z.); 2Department of Animal Science, Division of Agriculture, University of Arkansas, Fayetteville, NC 72701, USA

**Keywords:** antioxidative capacity, ferroptosis, gene expression, histological structure, intestinal mucosa, oxidative stress, polyphenols, weanling piglets

## Abstract

Polyphenols sourced from *Ilex latifolia* Thunb. (PIT) contain high levels of phenolic acids, tannic acids, triterpenoids and so on, which play important roles in antioxidant function. This study was conducted to investigate the effects of PIT against intestinal injury in piglets under oxidative stress. Thirty-two weanling piglets were arranged by a 2 × 2 factorial experiment with diets (basal diet vs. PIT diet) and oxidative stress (saline vs. diquat). All piglets were injected with saline or diquat on d 21, respectively. After 7 days, all pigs were slaughtered and intestinal samples were collected. PIT enhanced jejunal villus heights and crypt depth in the piglets under oxidative stress. PIT increased the activities of intestinal mucosal lactase, sucrase and maltase in the challenged piglets. PIT also increased the jejunal ratio of protein to DNA and ileal protein content. PIT increased the jejunal activities of GSH-PX and GSH content and reduced the ileal MDA amounts. Furthermore, PIT regulated the expression of ferroptosis mediators, such as *TFR1*, *HSPB1*, *SLC7A11* and *GPX4*. These results indicate that dietary PIT supplementation enhances the histological structure and function of the intestinal mucosa, which is involved in modulating antioxidant capacity and ferroptosis.

## 1. Introduction

In the current intensive swine production, changes in diet ingredients, contamination of mycotoxins in feeds, use of drugs and vaccines and other factors may lead to an excessive production of reactive oxide species (ROS), which cause oxidative stress in pigs [1]. Severe oxidative stress can induce tissue injury, especially intestinal injury [2]. Intestinal epithelial cells are rich in mitochondria, which are the main sites of ROS production [3]. ROS not only induce apoptosis and inhibit cell proliferation, but also inhibit intestinal development and interfere with intestinal function [4,5]. One study showed that oxidative stress led to histological damage in the jejunum with increased malondialdehyde and endotoxin concentration in piglets [6]. Moreover, Cao et al. (2020) demonstrated that oxidative stress resulted in intestinal epithelial barrier injury and mitochondrial damage in porcine intestinal epithelial cells [7]. Therefore, it is essential to alleviate intestinal injury caused by oxidative stress via nutritional regulation.

Ferroptosis is an LA test that identifies the type of cell death, which is closely related to oxidative stress in recent years [8,9]. The main characteristics of ferroptosis are the weakened repair ability of glutathione peroxidase 4 (GPX4) for lipid peroxidation injury, the accumulation of iron ions in cells, and the oxidation of polyunsaturated fatty acids containing phospholipids [10]. In terms of morphology, ferroptosis cells have characteristics, such as cell membrane integrity destroyed, mitochondrial cristae reduction or disappearance, and mitochondrial outer membrane rupture [8]. In terms of biochemistry, ferroptosis can result in the depletion of glutathione and a decrease in GPX4 activity [4].

*Ilex latifolia* Thunb. is called Da Ye Dong Qing in Chinese and is widely consumed in China and other Southeast Asia countries [11]. Polyphenols sourced from *Ilex latifolia* Thunb. (PIT) are a series type of plant polyphenols. In recent years, plant polyphenols in fruits, vegetables and seeds have been extensively studied for their excellent antioxidant and antibacterial abilities [12]. Furthermore, it has been found that the polyphenol extracts of beans, which are rich in tannic acids, have the ability to inhibit the growth of bacteria, fungi and yeast [13]. It is reported that supplementation with polyphenol complex in the diets of weanling piglets improved the antioxidant capacity and alleviated intestinal injury caused by *E. coli* stimulation [14]. Our lab has studied the protective effects of PIT on weanling piglets and obtained a series of findings. We found that PIT can alleviate intestinal inflammation and alter the microbiota composition in LPS-challenged piglets [15]. Moreover, PIT has a protective effect on hepatic damage in piglets under oxidative stress [16]. However, there are few reports on the effects of PIT on intestinal injury induced by oxidative stress in weanling piglets.

In this study, the weanling piglets were fed a basal diet with or without PIT, followed by an intraperitoneal injection of diquat to trigger intestinal oxidative stress and injury. The piglet challenged with diquat was a common method to establish an oxidative stress model [16]. This study aimed to explore whether PIT could improve intestinal health by regulating antioxidative capacity and the ferroptosis signaling pathway in the intestinal mucosa of piglets.

## 2. Materials and Methods

### 2.1. Experimental Animals and Design

The animal trial was conducted according to the Animal Scientific Procedures Act 1986 (Home Office Code of Practice. HMSO: London January 1997) and EU regulations (Directive 2010/63/EU). The whole procedure was approved by the Animal Care and Use Committee of Wuhan Polytechnic University (Wuhan, China). A total of 32 weanling piglets (Duroc × Landrace × Large White, with an age of 35 ± 1 d, and initial body weight (BW) of 8.16 ± 0.68 kg) were used in this experiment. Piglets were individually allotted in stainless steel metabolic cages (1.80 × 1.10 m^2^) with free access to feed and water in an environmentally controlled house. The experimental basal diet was formulated (Table 1) according to the National Research Council requirements (2012). A commercial polyphenols product, extracted from *Ilex latifolia* Thunb. (65.5% of the total polyphenols, mainly including phenolic acids and tannins, were analyzed by high-performance liquid chromatography), was supplemented with or without 250 mg/kg in the basal diet.

This experiment was designed with a 2 × 2 factorial trial. All pigs were fed a basal or PIT diet for 21 d and then intraperitoneally injected with diquat (dibromide monohydrate, Chem Service, West Chester, PA, USA) at the dose of 10 mg/kg BW in saline or the same volume of saline, respectively. The treatment factors were diet type (basal or PIT diet) and oxidative stress (diquat or saline).

### 2.2. Sample Collection

One week after the injection of diquat or saline solution, all piglets were humanely killed by intramuscular injection of sodium pentobarbital (80 mg/kg bodyweight). The 3-cm and 10-cm segments were cut from the mid-jejunum and mid-ileum in accordance with our previous study [17]. The 3-cm intestinal segments were gently flushed and stored in fresh 4% paraformaldehyde/PBS for histological analysis [18]. The 10-cm intestinal samples were opened longitudinally and flushed gently to remove luminal chyme. The mucosa samples were collected by scraping with sterile glass slides, then rapidly frozen in liquid nitrogen and stored at −80 °C for measurement of disaccharidase activities, contents of protein, DNA, RNA, antioxidase activities and mRNA and protein expression levels.

### 2.3. Intestinal Mucosal Histology

After a 24 h fixation, the intestinal segments were dehydrated, embedded, and stained with hematoxylin and eosin. Villus height and crypt depth were measured at 200× magnification with a microscope (Olympus CX31, Tokyo, Japan) according to our previous study [19]. Ten well-oriented and intact villi were selected and determined using a light microscope with a computer-assisted morphometric system (BioScan Optimetric; BioScan Inc., Edmond, WA, USA). Villus height was measured from the tip of the villus to the villus-crypt junction; crypt depth was defined as the depth of the invagination between adjacent villi.

### 2.4. Disaccharidases Activities of the Intestinal Mucosa

Disaccharidase activities in the intestinal mucosa were determined in accordance with our previous study using glucose kits (No. A082-1 for lactase, No. A082-2 for sucrase and No. A082-3 for maltase; Nanjing Jiancheng Bioengineering Institute, Nanjing, China) [18]. Briefly, 10 μL of double-distilled water, glucose standard solution (5.5 mmol/L) or test samples were added to a test tube and incubated with 20 μL of respective substrate for 20 min at 37 °C. Then, 10 μL of terminating agent and 1000 μL of a chromogenic agent were added and incubated at 37°C for 15 min. Double-distilled water was used to set zero at 505 nm, followed by the reading at the optical density value of each tube. One unit (U) of enzyme activity was defined as 1 nmol substrate hydrolysed/min under assay conditions (37 °C, pH 6.0).

### 2.5. Protein, DNA and RNA Contents of the Intestinal Mucosa

Frozen mucosal samples were homogenized in ice-cold NaCl solution at a 1:10 (*w*/*v*) ratio, followed by centrifugation at 2500 rpm for 10 min to collect the supernatant. The supernatant was used for the measurement of protein, DNA and RNA contents. Protein contents were measured according to the method of Lowry et al. [20]. DNA contents were measured by a fluorometric assay [21]. RNA contents were measured by spectrophotometry with a modified Schmidt–Tannhauser method [22].

### 2.6. Antioxidative Capacity of the Intestinal Mucosa

Frozen mucosal samples were pulverized in liquid nitrogen and homogenized in saline, then centrifuged at 2500 rpm for 10 min to acquire the supernatant. Total antioxidative capacity (T-AOC), activities of glutathione peroxidases (GSH-PX), contents of reductive glutathione (GSH) and malondialdehyde (MDA) of intestinal mucosa were determined by spectrophotometric methods following the instructions of the commercial kits’ manufacturer (Nanjing Jiancheng Bioengineering Institute, Nanjing, China).

### 2.7. Transmission Electron Microscope (TEM) Observation of the Intestinal Mucosa

The intestinal mucosa samples were dissected, fixed, dehydrated, sliced and stained in sequence. The intestinal mucosal slices were observed and photographed with an HT7700 TEM (Hitachi Co., Ltd., Tokyo, Japan) at an accelerating voltage of 80.0 kV and a magnification of 5000 in a blind manner.

### 2.8. Gene Expression Analysis

The procedure for total RNA isolation, quantification, reverse transcription, and real-time PCR were in accordance with previous study [19]. The primer pairs for amplification of target genes were shown in Table 2. The expression of the target genes relative to housekeeping gene (glyceraldehyde-3-phosphate dehydrogenase; GAPDH) was analyzed by the 2^−ΔΔCT^ method. Relative mRNA abundance of each target gene was normalized to the piglets fed basal diet and injected with saline.

### 2.9. Protein Abundance Analysis

The methods for protein abundance analysis in intestinal mucosa were referred to in previous methods [19]. In brief, the intestinal mucosa samples were homogenized in 600 μL of lysis buffer containing phenylmethanesulfonyl fluoride, protease and phosphatase inhibitors, and centrifuged at 12,000 *g* for 15 min at 4 °C to collect the supernatants. Equal amounts of intestinal mucosa protein were transferred onto 10–15% polyacrylamide gel and separated via SDS-PAGE, and then transferred to polyvinylidene difluoride membranes for immunoblotting. Immunoblots were blocked with 5% nonfat milk in Tris-buffered saline/Tween−20 for 3 h at room temperature (21–25 °C). The membranes were incubated overnight at 4 °C with primary antibodies, and then with the second antibodies for 2 h at room temperature. Specific primary antibodies included rabbit anti-transferrin receptor protein 1 (TFR1, 1:1000; 86 kDa, #70R-50471; Fitzgerald, Rd. Sudbury, Acton, MA, USA), goat anti-solute carrier family 7 member 11 (SLC7A11, 1:1000; 55 kDa, #ab60171; Abcam, Cambridge, MA, USA), rabbit anti-glutathione peroxidase 4 (GPX4, 1:1000; 20 kDa, #10005258; Cayman Chemical Company, Rd. Ellsworth, Ann Arbor, MI, USA) and mouse anti-β-actin antibody (1:1000, 43 kDa, #A2228; Sigma-Aldrich, St. Louis, MO, USA). Blots were developed using an Enhanced Chemiluminescence Western blotting kit (Amersham Biosciences, Solna, Sweden), and visualized using a Gene Genome bioimaging system. Brands were analyzed by densitometry using Gene Tools software (Syngene, Frederick, MD, USA). The relative protein abundance of target proteins (TFR1, SLC7A11, GPX4) was expressed as the ratio of target protein/β-actin protein.

### 2.10. Statistical Analyses

All data were analyzed as a 2 × 2 factorial experiment by ANOVA using the general linear model procedures (GLM) of SAS (SAS Inst. Inc., Cary, NC, USA). The statistical model included the effects of the diet type (basal diet or PIT diet), oxidative stress (saline or diquat) and their interactions. Data were presented as means and SEMs. When there was a significant interaction between diet and stress or a trend interaction between diet and stress, post hoc testing was conducted using Duncan’s multiple comparison tests. Differences were considered to be significant if *p* < 0.05.

## 3. Results

### 3.1. Intestinal Mucosal Histology

As shown in Table 3, there was a significant interaction between diet and stress on the jejunal villus height and crypt depth (*p* < 0.05). Supplementation with PIT in the diet significantly increased the jejunal villus height and crypt depth in the piglets under oxidative stress (*p* < 0.05). Oxidative stress significantly reduced the ratio of jejunal villus height to crypt depth (*p* < 0.05) and decreased the ileal villus height and crypt depth (*p* < 0.05). Compared with the piglets fed on a basal diet, the piglets fed on the PIT diet had significantly increased villus height, and the ratio of villus height to crypt depth in the ileum (*p* < 0.05). Similar to the above results, the histological appearance showed that supplementation with PIT in the diet alleviated intestinal mucosal injury of the piglets under oxidative stress (Figure 1 and Figure 2).

### 3.2. Disaccharidases Activities of the Intestinal Mucosa

There was a significant interaction between diet and stress on jejunal sucrase and maltase activities (*p* < 0.05, Table 4). Supplementation with PIT in the diet significantly increased the activities of jejunal sucrase and maltase in the piglets under oxidative stress (*p* < 0.05). Similarly, there was a significant interaction between diet and stress on ileal lactase, sucrase and maltase activities (*p* < 0.05). PIT significantly improved ileal lactase, sucrase and maltase activities in the piglets under oxidative stress (*p* < 0.05).

### 3.3. Protein, DNA and RNA Contents of the Intestinal Mucosa

There were significant interactions between diet and stress on the jejunal ratio of protein to DNA and ileal protein content (*p* < 0.05, Table 5). For the piglets injected with diquat, PIT significantly increased the jejunal ratio of protein to DNA and ileal protein content (*p* < 0.05). Furthermore, the piglets injected with diquat had a significantly reduced jejunal protein content and ileal ratio of RNA to DNA ratio (*p* < 0.05). Compared with the piglets fed the basal diet, the piglets fed the PIT diet had a significantly increased jejunal protein content and ileal ratio of RNA to DNA (*p* < 0.05).

### 3.4. Antioxidative Capacity of the Intestinal Mucosa

There were significant interactions between diet and stress on the activities of GSH-PX and GSH content in the jejunum (*p* < 0.05, Table 6). For the piglets under oxidative stress, PIT significantly increased the activities of GSH-PX and GSH content in the jejunum (*p* < 0.05). Moreover, the piglets injected with diquat had significantly reduced T-AOC and increased MDA content in the jejunum (*p* < 0.05). On the contrary, the piglets fed the PIT diet had significantly increased T-AOC and reduced MDA content compared with the piglets fed the basal diet (*p* < 0.05). There were significant interactions between diet and stress on the GSH and MDA amounts in the ileum (*p* < 0.05). For the piglets under oxidative stress, PIT significantly increased the GSH and reduced the MDA amounts in the ileum (*p* < 0.05). The piglets injected with diquat had significantly reduced T-AOC and GSH-PX activities in the ileum (*p* < 0.05). On the contrary, the piglets fed the PIT diet had significantly increased T-AOC and GSH-PX activities compared with the piglets fed the basal diet (*p* < 0.05).

### 3.5. TEM Observation of the Intestinal Mucosa

There were no obvious characteristics of ferroptosis in the intestinal epithelial cells of the piglets without oxidative stress (Figure 3A,B). However, characteristics of ferroptosis, such as the sparsely arranged microvilli, indistinct tight junction, disconnected tight junction, mitochondrial pyknosis, mitochondrial cristae reduction and dilatations of rough endoplasmic reticulum were observed in the piglets fed the basal diet under oxidative stress (Figure 3C). Interestingly, supplementation with PIT in the diet attenuated epithelial cells ferroptosis in the piglets under oxidative stress (Figure 3D).

### 3.6. Intestinal Mucosal Gene Expressions of the Key Genes Related to Ferroptosis

There were significant interactions between diet and stress on gene expressions of jejunal TFR1, SLC7A11 and GPX4 (*p* < 0.05, Table 7). Supplementation with PIT significantly reduced jejunal TFR1 gene expressions and improved SLC7A11 and GPX4 gene expressions in the piglets under oxidative stress (*p* < 0.05). Meanwhile, significant interactions between diet and stress on gene expressions of ileal TFR1, HSPB1 and GPX4 were observed (*p* < 0.05). Supplementation with PIT significantly reduced ileal TFR1 and HSPB1 gene expressions and improved GPX4 gene expressions in the piglets under oxidative stress (*p* < 0.05). Moreover, supplementation with PIT significantly increased ileal SLC7A11 gene expressions in the piglets under oxidative stress (*p* < 0.05).

### 3.7. Intestinal Mucosal Protein Abundance of the Key Proteins Related to Ferroptosis

There were significant interactions between diet and stress on jejunal TFR1 and GPX4 protein abundance (*p* < 0.05, Figure 4A). Supplementation with PIT significantly reduced TFR1 abundance while it increased GPX4 abundance in the jejunum of the piglets under oxidative stress (*p* < 0.05). Moreover, a significant interaction between diet and stress on ileal TFR1 protein abundance was observed (*p* < 0.05, Figure 4B). Supplementation with PIT significantly reduced TFR1 abundance in the ileum of the piglets under oxidative stress (*p* < 0.05). Furthermore, the piglets under oxidative stress had significantly reduced ileal SLC7A11 abundance (*p* < 0.05) and PIT supplementation significantly enhanced ileal SLC7A11 and GPX4 abundance (*p* < 0.05).

## 4. Discussion

An oxidative stress model of weanling piglets induced by diquat was established in this study, which is a mature and widely used method in animal experiments [23,24]. It has been reported that diquat injection could cause oxidative injury and impair intestinal absorption function in weanling piglets [25]. It is found that diquat-induced oxidative stress could damage the intestinal barrier function of piglets, with jejunal mucosal mitochondrial dysfunction and mitochondrial autophagy [25]. Plant polyphenols, as secondary metabolites of plants, have been proved useful in terms of antioxidant, anti-inflammatory and antiviral effects [26,27]. Therefore, this study was conducted to investigate whether PIT could alleviate diquat-induced intestinal injury in weanling piglets. In this study, it was found that supplementation with PIT improved intestinal mucosal histology and function, and enhanced the antioxidant capacity of the intestinal mucosa. In addition, PIT supplementation relieved the extent of intestinal epithelial cells’ ferroptosis by regulating the expression of genes and proteins related to ferroptosis.

Intestinal integrity is an important basis for assessing intestinal health. Intestinal integrity can be measured by a series of indicators, such as villus height and crypt depth, disaccharides activities, mucosal protein and DNA and RNA contents [28]. Villus height and crypt depth are the most intuitive indicators to reflect the morphological and structural integrity of intestinal mucosa [29]. Maltase, sucrase and lactase are disaccharides widely secreted in the intestinal tract. Disaccharides are involved in energy metabolism and are often used to measure digestive function [19]. Protein, DNA and RNA contents are important indicators of intestinal mucosa growth and development level, as well as injury repair status [30]. The ratio of RNA/DNA and protein/DNA can reflect mucosal protein synthesis level [31]. In this study, diquat injection reduced villus height and disaccharides activities, suggesting that diquat induced intestinal structural and functional impairment, which is consistent with previous studies [24,32]. PIT enhanced intestinal villus height, disaccharides activities, protein contents and the ratio of protein/DNA, which is in agreement with previous research [15]. Similar to our results, some studies found that polyphenols extracted from grape seeds or grape residue could increase the ratio of villus height/crypt depth, reduce the expression of pro-inflammatory factors, and improve digestion and absorption function in the intestine of pigs [33,34].

Intestinal injury is closely related to oxidative stress, which is caused by the imbalance of ROS amounts between production and elimination. ROS can damage cellular components, including lipids, DNA, proteins and carbohydrates, leading to tissue injury [35]. In the present study, diquat challenged decreased intestinal mucosal T-AOC, GSH-PX activities, and GSH contents, while increasing MDA contents, indicating that diquat successfully induced intestinal mucosal oxidative injury in piglets. Interestingly, supplementation with PIT mitigated these series of oxidative injuries. The phenolic hydroxyl structure of polyphenols is easily oxidized into the quinone structure, which consumes oxygen and captures ROS, causing polyphenols to have a strong antioxidant function [36]. Furthermore, it has been reported that polyphenols sourced from sorghum could maintain the balance between oxidants and antioxidants and play a role in alleviating oxidative stress [37]. Several swine nutrition studies have reported that polyphenol-rich diets could improve antioxidant status and reduce ROS levels [38,39,40]. Dietary chlorogenic acid supplementation improved the activities of GSH-PX and catalase in plasma and promoted growth performance by improving the antioxidant capacity of weanling piglets [41]. It was found that dietary catechin increased SOD activities and reduced H_2_O_2_ and MDA contents in the serum of pregnant sows [42]. Furthermore, it was reported that polyphenols in apples, grape seeds, green tea and olive leaves effectively improved the antioxidant capacity of weanling piglets, and reduced infections caused by *E. coli* [14]. The above results parameters demonstrated that PIT played a positive role in protecting intestinal histological injury and functional disorder of weanling piglets under oxidative stress. Although we determined the productive performance during this study, no significant difference was observed among these treatments. The current animal sample size was too small to get an accurate and productive performance. Maybe our next trial will explore the practical applicability of PIT by employing a large samples animal trial to determine a productive performance.

Dixon et al. (2012) found a new non-apoptotic mode of cell death driven by lipid peroxidation, which required intracellular enrichment of available ferrous ions, and associated this cell death with ferroptosis [8]. Studies have shown that tissue injury, caused by oxidative stress, is closely associated with ferroptosis [43]. Lipid peroxidation is the major feature of ferroptosis, and the organelle lesions of ferroptosis are represented by mitochondrial pyknosis, mitochondrial outer membrane rupture, mitochondrial cristae reduction and so on [10,44]. In the present experiment, we observed that the diquat injection can cause mitochondrial pyknosis, mitochondrial cristae reduction and dilatations of rough endoplasmic reticulum in the intestinal epithelial cells of piglets fed a basal diet under oxidative stress. However, dietary PIT could significantly alleviate organelle injury to a certain extent. These results suggest that diquat-induced oxidative stress might cause ferroptosis in intestinal epithelial cells, and PIT had protective effects on intestinal epithelial cells of piglets by alleviating ferroptosis.

Ferroptosis can be activated by some intracellular as well as extracellular factors. TFR1 is a receptor protein encoded by the transferrin receptor gene [45]. This protein can be used as a carrier to transfer ferric iron into the inner cell membrane when ferroptosis occurs. HSPB1 is a chaperone of the small heat shock protein (sHsp) group and it can reduce the contents of ferric iron by inhibiting the expression of TFR1, further alleviating ferroptosis [46]. The SLC7A11 gene codes for a sodium-independent cystine-glutamate antiporter, which is chloride dependent. As a component of the cysteine-glutamate transporter, SLC7A11 plays a key role in GSH homeostasis, which protects cells from oxidative injury [47]. GPX4 is a phospholipid hydroperoxidase which protects cells from membrane lipid peroxidation, and it can specifically inhibit ferroptosis [48]. In this study, the gene expressions of TFR1, HSPB1 and GPX4 increased after the diquat challenge, indicating that diquat induced large amounts of ferric iron into intestinal epithelial cells to cause oxidative stress. Meanwhile, the self-protection of the antioxidant system may be triggered as an explanation for the increased gene expressions of HSPB1 and GPX4. In addition, dietary PIT reduced the gene expressions of TFR1 and HSPB1 and increased the gene expressions of SLC7A11 and GPX4. Similar to the gene expression results, the protein abundance results also showed that supplementation with PIT enhanced GPX4 and SLC7A11 and decreased TFR1 protein abundance. These genes and protein expression results suggested that PIT could alleviate ferroptosis by inhibiting ferric iron transport and enhancing intestinal antioxidant capacity, which is in agreement with previous studies [16].

## 5. Conclusions

In conclusion, supplementation with PIT can alleviate diquat-induced intestinal mucosal histological and functional injury in the weanling piglets model. PIT can relieve diquat-induced intestinal ferroptosis by inhibiting the transfer of ferric iron and enhancing antioxidant capacity.

## Figures and Tables

**Figure 1 antioxidants-11-00966-f001:**
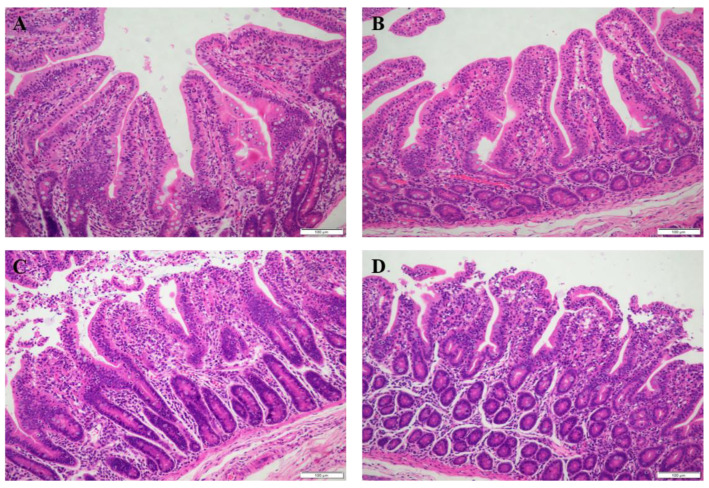
Jejunal mucosal histological appearance (hematoxylin and eosin) of the piglets fed polyphenols sourced from *Ilex latifolia* Thunb. (PIT) diets under oxidative stress. Original magnification 200×. Scale bars = 100 μm. (**A**) Piglets fed the basal diet and treated with saline. (**B**) Piglets fed the PIT diet and treated with saline. (**C**) Piglets fed the same basal diet and treated with diquat. (**D**) Piglets fed the same PIT diet and treated with diquat.

**Figure 2 antioxidants-11-00966-f002:**
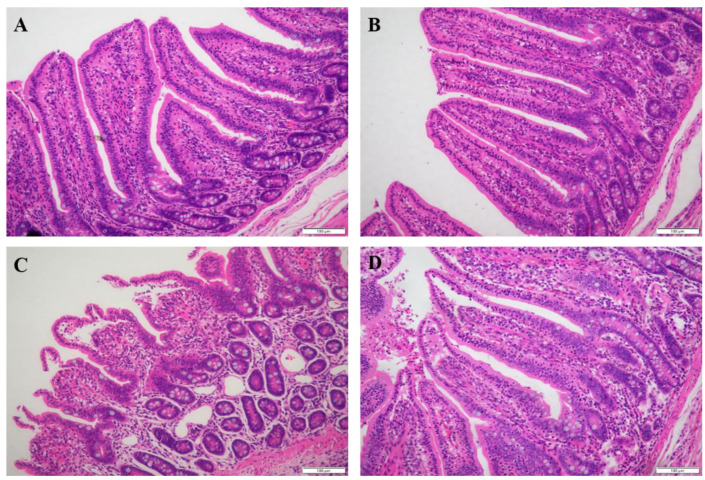
Ileal mucosal histological appearance (hematoxylin and eosin) of the piglets fed polyphenols sourced from *Ilex latifolia* Thunb. (PIT) diets under oxidative stress. Original magnification 200×. Scale bars = 100 μm. (**A**) Piglets fed the basal diet and treated with saline. (**B**) Piglets fed the PIT diet and treated with saline. (**C**) Piglets fed the same basal diet and treated with diquat. (**D**) Piglets fed the same PIT diet and treated with diquat.

**Figure 3 antioxidants-11-00966-f003:**
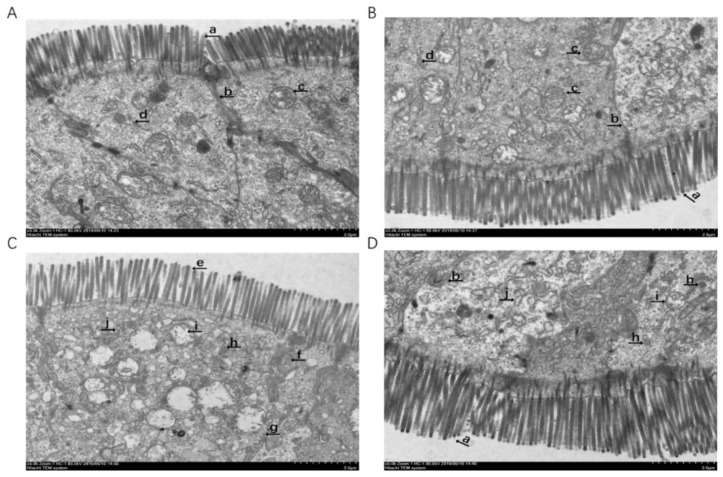
The epithelial cells ultrastructure in the jejunum of the piglets fed polyphenols sourced from *Ilex latifolia* Thunb. (PIT) diets under oxidative stress. Representative ultrastructure is shown. These pictures were obtained by transmission electron microscopy. (**A**) Piglets fed the basal diet and treated with saline. (**B**) Piglets fed the PIT diet and treated with saline. (**A**,**B**) There are no obvious ferroptosis characteristics. Presented as the closely arranged microvilli (a), normal tight junction (b), mitochondria with distinct cristae (c), normal rough endoplasmic reticulum (d). (**C**) Piglets fed the same basal diet and treated with diquat. Significant ferroptosis characteristics were observed, such as the sparsely arranged microvilli (e), indistinct tight junction (f), disconnected tight junction (g), mitochondrial pyknosis (h), mitochondrial cristae reduction (i) and dilatations of rough endoplasmic reticulum (j) were found. (**D**) Piglets fed the same PIT diet and treated with diquat. Original magnifications 5000×. Scale bars = 2 μm.

**Figure 4 antioxidants-11-00966-f004:**
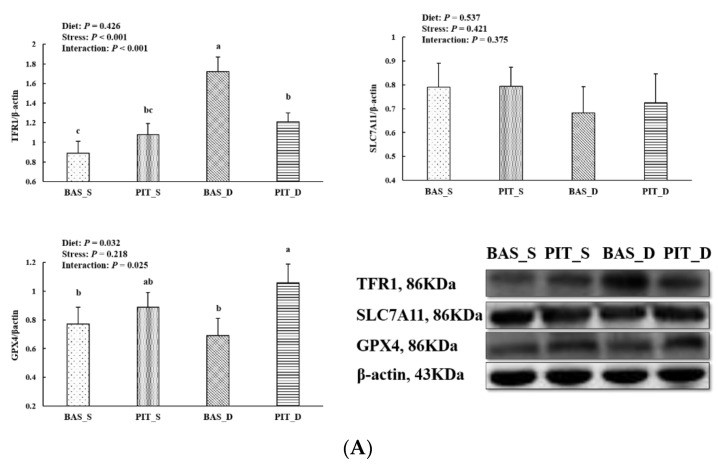
The jejunal (**A**) and ileal (**B**) mucosal protein abundance of ferroptosis-related signals of the piglets fed polyphenols sourced from *Ilex latifolia* Thunb. (PIT) diets under oxidative stress. The bands were the representative Western blot images. Values are mean and pooled SEM, *n* = 8 (1 piglet per pen). BAS_S, piglets fed the basal diet and injected with saline; PIT_S, piglets fed the PIT diet and injected with saline; BAS_D, piglets fed the basal diet and challenged with diquat; PIT_D, piglets fed the PIT diet and challenged with diquat. Different letters are significantly different between the treatment groups.

**Table 1 antioxidants-11-00966-t001:** Ingredient composition of experimental diets (%, as-fed basis).

Ingredients		Nutrients Level ^2^	
Corn	53.38	Digestible energy (MJ/kg)	14.82
Soybean meal, 44% crude protein	14.55	Crude protein	23.71
Fermented soybean meal	15.00	Calcium	0.80
Fish meal	6.00	Total phosphorus	0.63
Whey powder	5.00	Apparent total tract digestible phosphorus	0.36
Glucose	2.00	Total Lysine	1.59
Soybean oil	1.01	Standardized ileal digestible Lysine	1.35
Dicalcium phosphate	0.37	Total Methionine	0.50
Limestone	0.88	Standardized ileal digestible Methionine	0.44
Salt	0.30	Total Methionine + Cystine	0.86
L-Lysine HCl, 78%	0.34	Standardized ileal digestible Methionine + Cystine	0.72
L-Methionine, 98%	0.09	Total Threonine	0.99
L-Threonine, 98%	0.08	Standardized ileal digestible Threonine	0.79
Vitamin and mineral premix ^1^	1.00	Total Tryptophan	0.27
		Standardized ileal digestible Tryptophan	0.22

^1^ Premix supplied per kg diet: retinyl acetate, 5512 IU; cholecalciferol, 2200 IU; DL-α-tocopheryl acetate, 30 IU; menadione sodium bisulfite complex, 4 mg; riboflavin, 5.22 mg; D-calcium-pantothenate, 20 mg; niacin, 26 mg; vitamin B_12_, 0.01 mg; Mn (MnSO_4_·H_2_O), 40 mg; Fe (FeSO_4_·H_2_O), 75 mg; Zn (ZnSO_4_·7H_2_O), 75 mg; Cu (CuSO_4_·5H_2_O), 100 mg; I (CaI_2_), 0.3 mg; Se (Na_2_SeO_3_), 0.3 mg. ^2^ The nutrients level was analyzed value, except for digestible energy, apparent total tract digestible phosphorus, standardized ileal digestible Lysine, Methionine, Methionine + Cystine, Threonine and Tryptophan, which are calculated values.

**Table 2 antioxidants-11-00966-t002:** Primer sequence.

Name	Forward (5′–3′)	Reverse (5′–3′)	Annealing Temperature (°C)	Size (bp)	Accession Numbers
TFR1	CGAAGTGGCTGGTCATCT	TGTCTCTTGTCTCTACATTCCT	60	231	NM_214001.1
HSPB1	CTCGGAGATCCAGCAGACT	TCGTGCTTGCCCGTGAT	60	120	NM_001007518
SLC7A11	GCCTTGTCCTATGCTGAGTTG	GTTCCAGAATGTAGCGTCCAA	60	178	XM_021101587.1
GPX4	CTGTTCCGCCTGCTGAA	ACCTCCGTCTTGCCTCAT	60	218	NM_214407.1
GAPDH	CGTCCCTGAGACACGATGGT	GCCTTGACTGTGCCGTGGAAT	60	194	AF_017079.1

TFR1, transferrin receptor protein 1; HSPB1, heat shock protein beta 1; SLC7A11, solute carrier family 7 member 11; GPX4, glutathione peroxidase 4; GAPDH, glyceraldehyde-3-phosphate dehydrogenase.

**Table 3 antioxidants-11-00966-t003:** The intestinal mucosal histology of the piglets fed polyphenols sourced from *Ilex latifolia* Thunb. (PIT) diets under oxidative stress.

Items	Saline		Diquat		SEM	*p*-Value		
	Basal Diet	PIT Diet	Basal Diet	PIT Diet		Diet	Stress	Interaction
Jejunum								
Villus height (μm)	256 ^a^	271 ^a^	215 ^b^	267 ^a^	7	<0.001	<0.001	0.001
Crypt depth (μm)	164 ^b^	170 ^a,b^	147 ^c^	174 ^a^	4	<0.001	0.039	0.002
Villus height/crypt depth	1.58	1.60	1.47	1.54	0.04	0.085	0.003	0.354
Ileum								
Villus height (μm)	270	287	243	277	7	<0.001	0.001	0.107
Crypt depth (μm)	172	173	163	168	4	0.352	0.029	0.521
Villus height/crypt depth	1.58	1.67	1.49	1.66	0.05	<0.001	0.134	0.276

N = 8 (1 piglet per pen). The same letter on the shoulder of the mean in the same line indicates that the difference is insignificant, and the absence of the same letter means that the difference is significant. PIT, polyphenols sourced from *Ilex latifolia* Thunb.; SEM, standard error of mean.

**Table 4 antioxidants-11-00966-t004:** The activities of intestinal mucosal disaccharidases of the piglets fed polyphenols sourced from *Ilex latifolia* Thunb. (PIT) diets under oxidative stress (U/mg protein).

Items	Saline		Diquat		SEM	*p*-Value		
	Basal Diet	PIT Diet	Basal Diet	PIT Diet		Diet	Stress	Interaction
Jejunum								
Lactase	16.1	14.4	14.3	15.6	1.3	0.854	0.659	0.323
Sucrase	35.3 ^ab^	37.4 ^a^	28.6 ^b^	37.2 ^a^	3.5	0.025	0.410	0.013
Maltase	353 ^ab^	378 ^a^	301 ^b^	364 ^a^	27	0.129	0.048	0.035
Ileum								
Lactase	4.04 ^a^	3.89 ^ab^	2.81 ^b^	4.67 ^a^	0.58	0.241	0.488	0.026
Sucrase	65.2 ^a^	63.7 ^a^	50.1 ^b^	59.8 ^a^	4.4	0.157	0.036	0.004
Maltase	150 ^a^	163 ^a^	97 ^b^	143 ^a^	16	0.025	0.007	0.013

N = 8 (1 piglet per pen). The same letter on the shoulder of the mean in the same line indicates that the difference is insignificant, and the absence of the same letter means that the difference is significant. PIT, polyphenols sourced from *Ilex latifolia* Thunb.; SEM, standard error of mean.

**Table 5 antioxidants-11-00966-t005:** The contents of intestinal mucosal protein, DNA and RNA of the piglets fed polyphenols sourced from *Ilex latifolia* Thunb. (PIT) diets under oxidative stress.

Items	Saline		Diquat		SEM	*p*-Value		
	Basal Diet	PIT Diet	Basal Diet	PIT Diet		Diet	Stress	Interaction
Jejunum								
Protein (mg/g tissue)	5.48	5.89	4.87	5.13	0.22	0.021	0.017	0.674
RNA/DNA	6.14	6.57	5.87	5.93	0.53	0.376	0.078	0.488
Protein/DNA (mg/μg)	0.12 ^a^	0.14 ^a^	0.09 ^b^	0.14 ^a^	0.01	<0.001	0.653	0.042
Ileum								
Protein (mg/g tissue)	5.98 ^a^	6.03 ^a^	5.43 ^b^	5.94 ^a^	0.24	0.030	0.046	0.038
RNA/DNA	3.45	3.65	2.63	3.11	0.20	0.025	<0.001	0.388
Protein/DNA (mg/μg)	0.07	0.07	0.06	0.06	0.01	0.438	0.087	0.864

N = 8 (1 piglet per pen). The same letter on the shoulder of the mean in the same line indicates that the difference is insignificant, and the absence of the same letter means that the difference is significant. PIT, polyphenols sourced from *Ilex latifolia* Thunb.; SEM, standard error of mean.

**Table 6 antioxidants-11-00966-t006:** The intestinal mucosal antioxidative capacity of the piglets fed polyphenols sourced from *Ilex latifolia* Thunb. (PIT) diets under oxidative stress.

Items	Saline		Diquat		SEM	*p*-Value		
	Basal Diet	PIT Diet	Basal Diet	PIT Diet		Diet	Stress	Interaction
Jejunum								
T-AOC (U/mg protein)	0.543	0.665	0.472	0.532	0.046	0.019	0.032	0.564
GSH-PX (U/mg protein)	19.6 ^b^	28.6 ^a^	13.4 ^c^	29.7 ^a^	2	<0.001	0.243	0.036
GSH (mg GSH/g protein)	20.8 ^b^	25.1 ^a^	16.4 ^c^	26.0 ^a^	2.2	<0.001	0.351	0.013
MDA (nmol/mg protein)	1.43	1.21	2.46	1.66	0.28	0.017	<0.001	0.148
Ileum								
T-AOC (U/mg protein)	0.274	0.325	0.229	0.287	0.021	0.012	0.015	0.135
GSH-PX (U/mg protein)	21.2	30.0	14.3	26.5	1.9	<0.001	<0.001	0.795
GSH (mg GSH/g protein)	15.0 ^a^	15.1 ^a^	9.5 ^b^	13.4 ^a^	1.8	0.135	0.032	0.047
MDA (nmol/mg protein)	1.84 ^b^	1.73 ^b^	2.67 ^a^	1.98 ^b^	0.25	0.375	0.145	0.021

N = 8 (1 piglet per pen). The same letter on the shoulder of the mean in the same line indicates that the difference is insignificant, and the absence of the same letter means that the difference is significant. GPX4, glutathione peroxidase 4; GSH, glutathione; GSH-PX, glutathione peroxidases; MDA, malondialdehyde; PIT, polyphenols sourced from *Ilex latifolia* Thunb.; SEM, standard error of mean.

**Table 7 antioxidants-11-00966-t007:** The relative gene expressions of ferroptosis-related signals of the piglets fed polyphenols sourced from *Ilex latifolia* Thunb. (PIT) diets under oxidative stress.

Items	Saline		Diquat		SEM	*p*-Value		
	Basal Diet	PIT Diet	Basal Diet	PIT Diet		Diet	Stress	Interaction
Jejunum								
TFR1	1.00 ^b^	0.89 ^b^	1.67 ^a^	1.13 ^b^	0.15	0.022	0.015	0.034
HSPB1	1.00	0.94	0.88	0.99	0.10	0.781	0.641	0.332
SLC7A11	1.00 ^bc^	1.16 ^b^	0.75 ^c^	1.48 ^a^	0.17	0.035	0.743	<0.001
GPX4	1.00 ^c^	1.89 ^c^	5.35 ^b^	8.54 ^a^	0.51	<0.001	<0.001	0.015
Ileum								
TFR1	1.00 ^b^	0.92 ^b^	1.45 ^a^	0.94 ^b^	0.11	0.138	0.027	0.031
HSPB1	1.00 ^b^	0.76 ^c^	1.35 ^a^	0.83 ^bc^	0.11	<0.001	0.015	0.013
SLC7A11	1.00	1.46	0.31	0.68	0.13	0.019	<0.001	0.320
GPX4	1.00 ^b^	0.82 ^bc^	0.80 ^c^	1.43 ^a^	0.09	0.140	0.676	<0.001

N = 8 (1 piglet per pen). The same letter on the shoulder of the mean in the same line indicates that the difference is insignificant, and the absence of the same letter means that the difference is significant. GPX4, glutathione peroxidase 4; HSPB1, heat shock protein beta 1; PIT, polyphenols sourced from *Ilex latifolia* Thunb.; SEM, standard error of mean; SLC7A11, solute carrier family 7 member 11; TFR1, transferrin receptor protein 1.

## Data Availability

Data is contained within the article.

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
