# Peer review of "Polyphenols Sourced from Ilex latifolia Thunb. Relieve Intestinal Injury via Modulating Ferroptosis in Weanling Piglets under Oxidative Stress"

_antioxidants, 2022, doi:10.3390/antiox11050966_

Round 1

Reviewer 1 Report

The article corresponding to the ref.- antioxidants-1702111, entitled ”Polyphenols sourced from Ilex latifolia Thunb. Relieve intestinal injury via modulating ferroptosis in weanling piglets under oxidative stress” is intended to assess the polyphenols from Ilex latifolia to prevent intestinal injury induced by oxidative stress in piglets. In general, the article is well written and developed applying the sound scientific methodology, being of great interest to the research area. However, before its acceptance for publication, some issues need to be addressed. General comments Specific comments: ABSTRACT Some redundancies are found in this section, which is of special relevance given the limited number of characters/words in the section according to the guide for authors (lines 17-18/20-21). Also, it would be better to refer to the structural modification at the intestinal level as the histological structure of the intestinal mucosa more than morphology. In addition, although obviously the limited number of characters/words in the abstract does not allow provide many specific data, some quantitative results should be provided since the abstract is a section that should stand by itself. 5 out of the 6 keywords are already used in the title. I strongly recommend to the authors diversify the keywords to enhance the appearance of the article in the search of bibliographic databases. INTRODUCTION In page 1, lines 29-31, the authors state that beyond oxidative stress and induction of anergy, ROS also interfere with intestinal functions (include some examples of these effects). In page 1, line 41, the sentence “Plant polyphenols are secondary metabolites with polyphenol structure” is obviously redundant, redraft. The introduction does not clearly state the rationale behind the selection of PIT for studying the capacity of plant-extracts rich in polyphenols (among the broad diversity of plant-based foods featured by a high content of polyphenols) to prevent the tissue and functional damage at the intestinal level. Additional information is needed. In the last paragraph of the introduction, on page 2, an additional explanation of the aim and markers for monitoring the intestinal oxidative stress (as well as the selection criteria) have to be provided. MATERIAL AND METHODS In subsection 2.1, the formulation of the piglet’s diets is not clear at all. Does it mean that diets were supplemented to guarantee a concentration of 250 mg/kg of feed or polyphenolic ingests of 250 mg/kg of live piglet weight? Thereby, how much ingested per polyphenols per day each animal… Clarification is needed RESULTS AND DISCUSSION Figure 1C shows different magnifications relative to the images corresponding to additional treatments. So, the authors should correct the figure accordingly. In figure 2, SDSpage/WB should be enhanced by indicating the treatment corresponding to each column (although I assume that this should be the same shown in the bar plots that need to be specified). Finally, although all molecular markers analysed by the authors constitute valuable evidence of the biological effect of PIT in the frame of intestinal inflammation in piglets, in the end, these results should be interpreted based on productive parameters. I understand that the aim of the article is more related to biomolecular events but the relationship between both situations should be further discussed in the proper section resorting to data available in the literature to understand the practical applicability of the treatments assayed.

Reviewer 2 Report

The manuscript represents an interesting and well deigned study on weanling piglets aimed to evaluate anti-oxidant and cytoprotective effect of extract from Ilex latifolia Thunb. (PIT) in intestinum. Authors also examined modulation of ferroptosis, a novel recognized form of regulated cell death.  This report is sufficiently novel and results have practical application in pig breeding.

I have critical comments to several issues in this manuscript:

Parts  Introduction, Results and Discussion are described in sufficient detail, but part “ 2. Materials and Methods“ are elaborated vaguely. In general, when doing examinations of a basic research character such as this study, necessary information must be provided so that the study can be replicated by other experts.

2.1 Experimental Animals and Design.

Wy the dose of PIT 250 mg/kg was applicated. Is it recommended by producer? Was the consumption of provided food including PIT checked every day?

2.3 Intestinal Morphology.

Morphometry was performed on the sections of intestinal segments after staining. Authors should add representative images of the sections from each group of piglets to support numerical data. This is obvious in this kind of studies. How many morphometrical replicates from intestinal specimen prepared from each piglet/group was evaluated and used in statistical analysis? In the Legend of table 2 only this information is provided “N=8 (1 piglet per pen).“  Explain what means „piglet per pen“

Add to the Legend to Figure 1 that these pictures were obtained by Transmission Electron Microscopy.   

2.4 Disaccharidases Activities of The Intestinal Mucosa.

Authors refer to paper regarding determination of disaccharides:

  1. Liu, Y.L.; Huang, J.J.; Hou, Y.Q.; Zhu, H.L.; Zhao, S.J.; Ding, B.Y.; et al. Dietary arginine supplementation alleviates intestinal 374 mucosal disruption induced by Escherichia coli lipopolysaccharide in weaned pigs. Brit. J. Nutr. 2008, 100, 552-560.

However, in this study disaccharides in intestinal mucosa were not determined. Used kits are specific for this part of scientific community and it is necessary to provide basic information how disaccharides were determined and how were measured.

2.5 Protein, DNA and RNA Contents of The Intestinal Mucosa

Authors refer that “Protein, DNA and RNA contents were measured using the supernatant according to previous studies [16].

  1. Zhu, H.L.; Liu, Y.L.; Chen, S.K.; Wang, X.Y.; Pi, D.A.; Leng, W.B.; et al. Fish oil enhances intestinal barrier function and inhibits 379 corticotropin-releasing hormone/corticotropin-releasing hormone receptor 1 signalling pathway in weaned pigs after lipopolysaccharide challenge. Brit. J. Nutr. 2016, 115, 1947-195.

However, in this paper methodologies for measurement of these parameters are not described in details. There is only paragraph “Intestinal alkaline phosphatase activity analysis“ where there is mentioned that „The protein concentrations of intestinal mucosa were determined using Coomassie Brilliant Blue G-250 reagent with bovine serum albumin as a standard.„

There is missing information which solutions were used for extraction of proteins, DNA and RNA during homogenization. Is it correct that DNA and RNA was measured in supernatants and not in cells? Were supernatants centrifuged? Why you decided to measure proteins in supernatants? The proper description of methodology must be included.  

2.6 Anti-oxidative Capacity of The Intestinal Mucosa.

Authors refer to study:

  1. He, P.W.; Hua, H.W.; Tian, W.; Zhu, H.L.; Liu, Y.L.; Xu, X. Holly (Ilex latifolia Thunb.) polyphenols extracts alleviate hepatic damage by regulating ferroptosis following diquat challenge in a piglet model. Front. Nutr. 2020, 7, 604328.

However, in this study „Anti-Oxidative Capacity of the Liver“ is very briefly described. In present submitted MS „glutathione peroxidases (GSH-PX), contents of reductive glutathione (GSH) and malondialdehyde (MDA)“ were determined in intestinal mucosa using commercial kits. The concise description of mucosal sample preparation and methodology is necessary to allow reproducibility of results. These kits might not be available for many of scientists in the world.

2.9 Protein Abundance Analysis“ is actually  quantitative western blot analysis. Authors refer to study:

  1. Xu, X.; Chen, S.K.; Wang, H.B.; Tu, Z.X.; Wang, S.H.; Wang, X.Y.; et al. Medium-chain TAG improve intestinal integrity by suppressing toll-like receptor 4, nucleotide-binding oligomerisation domain proteins and necroptosis signalling in weanling piglets challenged with lipopolysaccharide. Brit. J. Nutr. 2018, 119, 1019-1028.

However in this paper there is used another citation in methodology paragraph „The analysis of protein abundance in intestinal mucosa was carried out according to the method described previously (27).“ How the proteins were isolated? What was the source of proteins, supernatants (used in previous analysis) or homogenized mucosal cells? What kind of membrane was used for blotting of proteins? I suppose that authors quantified bands intensity by densitometry, describe how.

2.10 Statistical Analyses.

It is not explained why SEM was calculated and not SD for each set of results for each experimental group. SD value is more specific and sensitive for statistical analysis,

table 6: “The intestinal mucosal gene expressions of ferroptosis-related signals of ...“ Authors analysed gene expression according the 2-△△CT method, therefore I suggest to specify more the numbers summarised is Table as „Fold change“ or „Relative gene expression“ of examined m-RNA level“.

Discussion:

Line 286: The phenolic hydroxyl structure of polyphenols is easily oxidized into quinone structure, which consumes oxygen and captures ROS, making polyphenols have strong anti-oxidant function [17]. This reference is not directly related to this statement.

Reviewer 3 Report

The present manuscript contains numerous methodological shortness which prevent me to accept it in present form.

  1. In general, methodology is very laconic. I consider it necessary to develop a more precise description of the methods used and the material used in the study. The authors should remember that a reader should be able to repeat their experiment.
  2. Has basal diet been checked for the presence of polyphenols ?
  3. The detailed composition of basal as well as PIT diet must be provided.
  4. The detailed description of piglet euthanasia is missing.
  5. Villus and crypts size might be measured with a software but only viewed with a microscope. Please add information what kind of software was used.
  6. Line 86 – those two sentences are not related to the intestine morphology but rather to the mucosa histology.
  7. Why only jejunum and ileum were dissected ? What about the duodenum? This must be clarified. By the way in the weaned piglets ileum is really short. So what was a reference point for mid-ileum?
  8. Tables and figures – what a, b and c stand for ?
  9. Illustration of HE stainings is missing.
  10. The conclusions are too speculative and simplistic. What do the authors mean by "well feed additives"? “Well” in what sense?

Round 2

Reviewer 3 Report

I can accept all explanations and corrections provided by the authors.